# Reaction of Picolinamides with Ketones Producing a New Type of Heterocyclic Salts with an Imidazolidin-4-One Ring

**DOI:** 10.3390/molecules29010206

**Published:** 2023-12-29

**Authors:** Eugenia P. Kramarova, Dmitry N. Lyakhmun, Dmitry V. Tarasenko, Sophia S. Borisevich, Edward M. Khamitov, Alfia R. Yusupova, Alexander A. Korlyukov, Alexander R. Romanenko, Tatiana A. Shmigol, Sergey Yu. Bylikin, Yuri I. Baukov, Vadim V. Negrebetsky

**Affiliations:** 1Institute of Pharmacy and Medical Chemistry, Pirogov Russian National Research Medical University, Ostrovityanov St., Bl. 1, 117997 Moscow, Russia; kramarova_ep@rsmu.ru (E.P.K.); liakhman_dn@rsmu.ru (D.N.L.); tarasenko_dv@rsmu.ru (D.V.T.); alex@xrlab.ineos.ac.ru (A.A.K.); shmigol_ta@rsmu.ru (T.A.S.); baukov@rsmu.ru (Y.I.B.); 2Ufa Institute of Chemistry, Oktyabrya Aven., 71, 450054 Ufa, Russia; ssborisevich@mephi.ru (S.S.B.); khamitovem@gmail.com (E.M.K.); alfia_yusupova@mail.ru (A.R.Y.); 3A.N.Nesmeyanov Institute of Organoelement Compounds of Russian Academy of Sciences, Vavilova St. 28, Bl. 1, 119334 Moscow, Russia; romanenko_ar@ineos.ac.ru; 4D.I. Mendeleev Russian University of Chemical Technology, Miusskaya Sq., 9, 125047 Moscow, Russia; 5The Open University, Walton Hall, Milton Keynes MK7 6BJ, UK

**Keywords:** imidazolidin-4-ones, sulfobetaines, NMR and FT-IR spectroscopy, X-ray study, quantum-chemical calculations

## Abstract

Reactions of picolinamides with 1,3-propanesultone in methanol followed by the treatment with ketones led to a series of previously unknown chemical transformations, yielding first pyridinium salts (**2a**–**f**), with a protonated endocyclic nitrogen atom, and then heterocyclic salts (**3a**–**j**) containing an imidazolidin-4-one ring. The structures of intermediate and final products were determined by IR and ^1^H, ^13^C NMR spectroscopy, and X-ray study. The effects of the ketone and alcohol structures on the product yield were studied by quantum-chemical calculations. The stability of salts **3a**–**j** towards hydrolysis and alcoholysis makes them excellent candidates for the search for new types of biologically active compounds.

## 1. Introduction

Sulfobetaines, commonly prepared by the reactions of tertiary and aromatic amines with sultones, are objects of intense studies because of their unusual reactivity and a broad spectrum of applications, including redox reagents [1], polymers, surfactants [2], and medical drugs [3,4,5,6,7,8].

In our previous works, traditional synthetic approaches [9] were used for the preparation of novel pyridinecarboxamides (**A**) containing a homotaurin fragment [10]. Typically, the reactions proceeded through the opening of the sultone ring, as shown in Figure 1.

For the reactions of 3- and 4-pyridinecarboxamides with sultones in boiling methanol, the position of the amido group has little effect on the yield of the final sulfobetaine **A**. In contrast, 2-pyridinecarboxamides (picolinamides) under similar conditions produce pyridinium salts **B** (Figure 2).

The low yield of product **A** in the case of picolinamides and the formation of salt **B** at high temperatures are likely to be a result of intramolecular bonding in the substrate involving the amido group and endocyclic nitrogen atom [10].

The optimisation of the reaction conditions unexpectedly yielded the previously unknown 3,3-dimethyl-1-oxo-2,3-dihydro-1*H*-imidazo [1,5-*a*]pyridin-4-ium 3-methoxypropane-1-sulfonate (**3a**) [11]. To the best of our knowledge, compound **3a** is the first example of imidazolidin-4-one salts.

According to literature, natural derivatives of imidazolidin-4-one, such as oxaline, neoxaline [12] and hetacillin [13], demonstrate a broad spectrum of antibacterial, antifungal and antitumour activity. Biologically active synthetic imidazolidin-4-ones include spiperone and mosapramine, which are potent dopamine receptor antagonists [14] and clinically important antipsychotic agents [15].

At present, imidazolidin-4-ones are usually prepared by multistage synthetic processes [16,17,18,19,20,21,22,23,24] (Figure 3).

In some cases, a protective group must be used. The synthesis of *N*-substituted target compounds requires additional chemical transformations, such as condensations of α-acetaminoamides with aldehydes or ketones [25,26,27]. Other common approaches to *N*-substituted substrates include the Curtius rearrangement [28,29,30], Buchwald–Hartwig amination [31], reductive amination of carbonyl compounds [32] and Mitsunobu reaction [33].

In this study, the synthesis, structure and properties of new types of imidazolidin-4-ones with novel organic anions are reported.

## 2. Results

### 2.1. Synthesis

In the present work, the reactions of picolinamide **1a** with various alcohols and ketones were explored. Under similar conditions, these reactions produced a broad range of pyridinium (**2a**–**g**) and imidazolidin-4-ones (**3a**–**j**) salts (Figure 4, Table 1 and Table 2, see also Experimental Part).

In the case of *N*-substituted amide **1b**, the intermediate pyridinium salt **2g** was found to be unreactive towards acetone under studied reaction conditions, and the formation of the corresponding imidazolidin-4-one derivative was not observed.

Our attempt to prepare compound **3a** by refluxing salt **2a** (which was isolated by the evaporation of the solvent and used without further purification from the traces of methanol) in acetone for 3 h was only partially successful, as the yield was impractically low (12%). However, when a hot solution of **2a** in methanol was treated with acetone, the yield increased to 80%. These results suggest that the formation of **3a** in the second case could involve the reaction of **2a** with a hemiketal intermediate (Figure 5).

Most compounds **3a**–**j** were obtained with high yields (75–94%, Table 2). The lower yields of compounds **3g** (43%) and **3i** (23%) could be caused by elimination reactions of alcohols used for the preparation of pyridinium salts **2c** and **2e**, respectively. The separation of **3i** and **2e** was very problematic, so the ^1^H NMR spectrum of the final mixture in D_2_O showed very broad signals of both compounds in 4:1 ratio, respectively (see Experimental Part). The IR spectrum of the mixture also showed the characteristic absorptions of both salts (see Experimental Part).

Our attempts to carry out the condensation of **2a** with methyl tert-butyl ketone, acetophenone and benzaldehyde were unsuccessful—in all cases, only the original salt was isolated.

### 2.2. X-ray Study

According to the results of the X-ray study, the values of all bond lengths and angles in salts **2a** and **3a** (Figure 1) fall within the ranges typical for pyridinium salts of alkylsulfonic acids. Crystallographic data for **2a** and **3a** are summarised in Table 3 (see Experimental Part). The parameters of hydrogen bonds are shown in Appendix A.

Salt **2a** crystallised in a monoclinic system and chiral space group P2_1_. The unique part of the unit cell contained four crystallographically independent cations and anions linked by strong hydrogen bonds between sulfo groups of anions and amido or pyridinium moieties of cations (Figure 2, left).

The supramolecular structure formed by hydrogen bonds can be described as a double layer, with ether groups of anions residing inside the layers, while sulfo groups of anions and pyridinium moieties of cations form the outer shell. In turn, the double layers are held together by weak interactions between sulfo groups and ipso-carbon atoms of pyridinium moieties.

In the crystal packing of **3a**, cations and anions form dimers via hydrogen bonds between the sulfo group of the anion and the imidazolidin-4-one ring of the cation. In turn, these dimers are assembled into a 3-D framework via weak C–H…O interactions (Figure 2, right).

### 2.3. Reactions of Hydrolysis of Compounds **3a**, **3d** and **3e**

Many derivatives of imidazolidin-4-one are unstable in acidic and neutral aqueous environments. For example, the half-life of the antibacterial drug hetacillin in aqueous solutions at pH 3–8 is approximately 30 min [34]. The main hydrolysis product, ampicillin, is responsible for over 90% of the biological activity of hetacillin [35]. The N’-alkylation improves the hydrolytic stability of imidazolidin-4-ones both in human plasma and aqueous buffer with pH 7.4 [19].

In order to evaluate the applicability of salts **3a**–**j** as potential drug candidates, we have studied the stability of compounds **3a**, **3d** and **3e** towards water.

In contrast to hetacillin, compound **3a** was stable in water at room temperature (no hydrolysis was observed over a period of seven days). The reflux of compound **3a** in water for 5 h led to the elimination of acetone and the formation of salt **2a** with a yield of 32% (Figure 2). The hydrolysis of the same compound at moderate temperatures (70–80 °C) increased the yield of **2a** to 63%.

A similar behaviour was observed for compounds **3d** and **3e**. For example, the reaction of **3d** with water at 70–80 °C for 5 h produced a mixture of **2a** and **3d**, with the IR spectrum showing two ν(C=O) bands at 1708 and 1727 cm^−1^, respectively.

### 2.4. Theoretical Study

Thermodynamic parameters of the chemical reactions shown in Figure 4 were estimated using quantum-chemical calculations.

#### 2.4.1. Thermodynamic Parameters of Formation for Compounds **2a**–**f**

The first step of the reaction leading to salts **2a**–**f** (Figure 3) is the nucleophilic addition of an alcohol to 1,3-propanesultone, which produces 3-alkoxypropanesulfonic acids **1-IIa**–**f** (Figure 6):

With the exception of R = i-Bu, the thermal effect of this step decreases when the size of the R substituent increases (Figure 3, Appendix A).

Therefore, sterical hindrance is likely to be the major factor affecting the concentrations of 3-alkoxypropanesulfonic acids **1-IIa**–**f** in the reaction mixture, which in turn affects the yields of respective salts **2a**–**f** (Figure 3).

#### 2.4.2. Structures of the Cation–Anion Complexes

The actual thermodynamic parameters of chemical reactions shown in Figure 3 depend on the mutual orientation of cations and anions under experimental conditions (in boiling methanol). Possible structures of salt **2a** in the reaction medium could be predicted by analysing electrostatic potential (ESP) maps of the constituent ions (Figure 4A).

The red and blue dots in Figure 4A correspond to regions of ESP maps with low and high electron density, respectively. The lowest electron density for **1-Ia** is observed near the nitrogen atom of the pyridinium ring, while the highest electron density is concentrated around the sulfo group of the anion **1-IIa**.

The most common mutual positions of ions **1-Ia** and **1-IIa** were identified by statistical analysis of molecular dynamics (MD) simulation for a system containing two ions of opposite charge and a large number of methanol molecules. This allowed us to take into account the solvation of salt **2a** by methanol and thus predict the most likely structure of the complex in solution (Figure 4B).

According to our calculations, salt **2a** in solution is stabilised by hydrogen bonds involving one of the oxygen atoms in anion **1-IIa** and hydrogen atoms at N1 and N2 in cation **1-Ia** (Figure 4B). As expected, the calculated geometric parameters of the cation–anion complex in solution differ significantly from those in the solid state obtained by X-ray study (Figure 1). The most obvious difference is the mutual orientation of the acetamide fragment and pyridinium ring in **1-Ia**. In solution (Figure 4B), the calculated value of the dihedral angle (φ) N1–C–C–N2 is close to zero while in the solid state (Figure 1) it approaches 180°. According to quantum-chemical calculations, the most stable conformation of non-protonated picolinamide **1a** is achieved at φ ≈ 0° [10]. However, this is not the case for protonated picolinamide in **2a**, where the conformation with φ ≈ 180° is by ca. 20 kJ/mol more stable (Figure 4C). The reason for this difference is the formation of hydrogen bonds: intramolecular in **1a** (φ ≈ 0°) and interionic in the **2a** complex (φ ≈ 180°). In the latter case, the AIM analysis reveals critical points of (3; −1) type (for more detail, see Appendix A).

The activation energy for the proton migration from sulfonic acid to the pyridine ring is relatively low (4.5 kJ/mol, M052X-D3/TZVP), which suggests that the formation of salt **2a** might involve the following transition state **C** (Figure 7).

#### 2.4.3. Thermodynamic Parameters of **2a**–**g** Formation

Figure 5 shows a histogram where the yields of salts **2a**–**f** are plotted together with relative differences of the reaction enthalpy and Gibbs free energy, with zero values for ΔΔ_r_H° and ΔΔ_r_G° corresponding to the lowest values of Δ_r_H° and Δ_r_G° from Appendix A.

According to Figure 5, relative values ΔΔ_r_H° and ΔΔ_r_G° generally increase along with the size of the alkyl chain in the alcohol. The only exception is isopropanol, which also has the lowest values of Δ_r_H° and Δ_r_G° for the reaction with 1,3-propanesultone (Figure 5, Appendix A). The least thermodynamically favourable reaction, the formation of salt **2e**, proceeds with the lowest yield (25%). At the same time, similar ΔΔ_r_H° and ΔΔ_r_G° values are observed for salt **2f**, which was obtained with the highest yield (89%). The formation of salt **2g** is characterised by positive absolute values of Δ_r_H° and Δ_r_G° (Appendix A).

#### 2.4.4. Thermodynamic Parameters of **3a**–**j** Formation

Analysis of thermodynamic parameters of salts **3a**–**j** formation suggests that the observed chemical changes could be divided into three groups. The first group includes the formation of salts **3a**–**c** by reactions of **2a** with aliphatic ketones (Figure 6A).

In this group, an increase in the alkyl substituent size raises the ΔΔ_r_H° and ΔΔ_r_G° values and, therefore, lowers the reaction yield. The absolute values of Δ_r_H° and Δ_r_G° are given in Appendix A.

The second group includes the reactions of salt **2a** with cyclic ketones. In this case, an increase in the Reactions of the third group includes the formation of salts **3a** and 3**f**–**j**. In these reactions, an increase in the size of the alkoxy substituent in the sulfonic acid raises the ΔΔ_r_H° and ΔΔ_r_G° values and, therefore, lowers the reaction yield (Figure 6B). These results correlate with the data shown in Figure 5. Ring size lowers the ΔΔ_r_H° and ΔΔ_r_G° values and, therefore, raises the reaction yield.

## 3. Materials and Methods

### 3.1. Chemistry

The purities of all compounds were assessed by elemental analysis and NMR and found to be ≥95%. NMR spectra were recorded on a Bruker Avance II 300 spectrometer (Bruker BioSpin GMBH, Rheinstetten, Germany) at 300 MHz (^1^H) and 75 MHz (^13^C) in D_2_O in the pulse mode, followed by Fourier transformation using Me_4_Si as the internal standard. Spin multiplicities are designated as s (singlet), d (doublet), t (triplet), q (quartet) or m (multiplet). IR spectra in the solid phase were recorded on a Bruker Tensor-27 instrument (Bruker Corporation, Bremen, Germany) with an attenuated total internal reflectance (ATR) module. Refraction parameters were measured using an IRF-454B2M refractometer (KOMZ, Kazan, Russia). Melting points were determined using a Stuart SMP10 instrument (Barloworld Scientific Ltd., Stone, UK). Elemental analyses were carried out at the Laboratory of Organic Microanalysis of INEOS RAS.

#### Synthesis

Compounds **1a**, **1b**, 1,3-propanesultone, ketones and alcohols were obtained as commercial reagents from Acros and Sigma–Aldrich and used without further purification.

General synthesis of compounds **2a**–**g** and **3a**–**i.** A mixture of 0.005 mol of compound **1a** or **1b** and 0.006 mol of 1,3-propanesultone in alcohol was refluxed for 4 h. The solvent was evaporated, and the salt **2a**–**f** obtained was treated with a ketone at reflux in methanol. The crystals of salt **3a**–**j** formed were filtered out and dried.

Synthesis of 2-carbamoylpyridin-1-ium 3-methoxypropane-1-sulfonate (**2a**): According to the general protocol (MeOH, 4 mL), 1.52 g (65%) of **2a** was obtained, m.p. 153–155 °C (acetonitrile). IR spectrum (solid, ν/cm^−1^): 1707 s (C=O), 1602 (C=C_pyridine_), 1179 s, 1124 s, 1035 s (SO_3_). ^1^H-NMR (300.1 MHz, D_2_O, δ, ppm, J/Hz): 1.88 (m, 2H, -H_2_C**CH_2_**CH_2_-), 2.81 (t, 2H, ^3^J = 6.5, -CH_2_SO_3_), 3.46 (t, 2H, ^3^J = 6.5, CH_3_O**CH_2_**CH_2_-), 3.23 (s, 3H, **CH_3_O**CH_2_), 8.35 (d, 1H, ^3^J = 7.9, H3), 8.08 (t, 1H, ^3^J = 7.9, H4), 8.58 (t, 1H, ^3^J = 7.9, H5), 8.77 (d, 1H, ^3^J = 7.9, H6); ^13^C-NMR (75.5 MHz, D_2_O, δ, ppm): 24.03, 47.88, 57.61, 70.47, 125.06, 129.57, 142.51, 143.21, 146.96, 162.39. Anal. calcd. for C_10_H_16_N_2_O_5_S: C, 43.47; H, 5.84; N, 10.14; S, 11.60; O, 28.95. Found: C, 42.85; H, 6.54; N, 9.09; S, 10.40; O, 31.13.

Synthesis of 2-carbamoylpyridine-1-ium-3-ethoxypropane-1-sulfonate (**2b**): A mixture of 0.61 g (0.005 mol) picolinamide **1a** and 0.73 g (0.006 mol) 1,3-propanesultone in 4 mL of ethanol was refluxed for 3 h, the volatiles were removed in a vacuum, and the residue was stirred with 10 mL of diethyl ether for 1 h. The crystals formed were isolated by filtration to yield 1.34 g (85%) of compound **2b ·** 1.5 H_2_O, m.p. 145–147 °C (CH_3_CN). IR spectrum (solid, ν/cm^−1^): 1707 s (C=O), 1602 w (C=C_pyridine_), 1176 s, 1146 s, 1034 s (SO_3_). ^1^H-NMR (300.1 MHz, D_2_O, δ, ppm, J/Hz): 1.09 (t, 3H, ^3^J = 7.0, **CH_3_**CH_2_-), 1.87 (m, 2H, -CH_2_**CH_2_**CH_2_-), 2.85 (t, 2H, ^3^J 7.2, -CH_2_SO_3_), 3.51 (m, 4H, -**CH_2_**O**CH_2_**-), 8.41 (d, 1H, ^3^J = 8.1, H3), 8.13 (t, 1H, ^3^J = 8.1, H4), 8.68 (t, 1H, ^3^J = 8.1, H5), 8.84 (d, 1H, ^3^J = 8.1, H6). ^13^C-NMR (75.5 MHz, D_2_O, δ, ppm): 14.14, 23.57, 47.99, 60.22, 72.29, 125.18, 129.68, 142.67, 143.38, 147.38, 162.58. Anal. calcd. for C_11_H_21_N_2_O_6.5_S: C, 41.63; H, 6.67; N, 8.83; S, 10.10. Found: C, 41.77; H, 6.35; N, 9.03; S, 9.61.

Synthesis of 2-carbamoylpyridine-1-ium-3-isopropoxypropane-1-sulfonate (**2c**): Similar to **2b**, using isopropanol instead of ethanol, 1.43 g (59%) of compound **2c ·** HO(CH_2_)_3_SO_3_H **·** CH_3_CN was obtained, m.p. 180–181 °C (CH_3_CN). IR spectrum (solid, ν/cm^−1^): 1708 s (C=O), 1637 w (C=C_pyridine_), 1176 s, 1146 s, 1036 s (SO_3_). ^1^H-NMR (300.1 MHz, D_2_O, δ, ppm, J/Hz): 1.05 (br. s, 6H, 2CH_3_), 3.60 (br. m, 1H, CH), 1.86 (m, 2H, -H_2_C**CH_2_**CH_2_-), 2.85 (br. m, 2H, ^3^J = 7.1, -CH_2_SO_3_), 3.50 (br. t, 2H, ^3^J = 7.1, OCH_2_-), 8.38 (d, 1H, ^3^J = 8.1, H3), 8.11 (t, 1H, ^3^J = 8.1, H4), 8.63 (t, 1H, ^3^J = 8.1, H5), 8.79 (d, 1H, ^3^J = 8.1, H6). ^13^C-NMR (75.5 MHz, D_2_O, δ, ppm): 21.15, 24.50, 48.01, 66.13, 72.28, 125.14, 129.66, 142.83, 143.54, 146.87, 162.79. Anal. calcd. for C_17_H_31_N_3_O_9_S_2_: C, 42.04; H, 6.43; N, 8.65; S, 13.21. Found (%): C, 41.91; H, 6.32; N, 8.85; S, 12.39.

Synthesis of 2-carbamoylpyridine-1-ium-3-isobutoxy propane-1-sulfonate (**2d**): Similar to **2b**, using isobutanol at 78–80 °C instead of ethanol at reflux, 0.98 g (57%) of compound **2d** · 1.5 H_2_O was obtained, m.p. 179–182 °C (CH_3_CN). IR spectrum (solid, ν/cm^−1^): 1708 s (C=O), 1601 w (C=C_pyridine_), 1146 s, 1124 s, 1036 s (SO_3_). ^1^H-NMR (300.1 MHz, D_2_O, δ, ppm, J/Hz): 0.83 (d, 6H, ^3^J = 7.1, 2CH_3_), 1.76 (m, 1H, CH), 1.96 (m, 2H, CH_2_**CH_2_**CH_2_), 2.91 (t, 2H, ^3^J = 7.1, CH_2_SO_3_), 3.55 (t, 2H, ^3^J = 7.1, OCH_2_), 3.23 (d, 2H, ^3^J = 7.1, -CH**CH_2_**O), 8.42 (d, 1H, ^3^J = 8.1, H3), 8.15 (t, 1H, ^3^J = 8.1, H4), 8.65 (t, 1H, ^3^J = 8.1, H5), 8.85 (d, 1H, ^3^J = 8.1, H6). ^13^C-NMR (75.5 MHz, D_2_O, δ, ppm): 18.57, 24.30, 27.55, 48.06, 68.81, 77.48, 125.14, 129.66, 142.83, 143.54, 146.87, 162.77. Anal. calcd. for C_13_H_25_N_2_O_6.5_S: C, 45.20; H, 7.29; N, 8.10; S, 9.28. Found: C, 45.72; H, 6.69; N, 8.31; S, 8.71.

Synthesis of 2-carbamoylpyridine-1-ium-3-tert butoxy propane-1-sulfonate (**2e**): Similar to **2b**, using tert-butanol instead of ethanol, 0.62 g (25%) of compound **2e** · 1.5 HO(CH_2_)_3_SO_3_H · CH_3_CN was obtained, m.p. 149–150 °C (CH_3_CN). IR spectrum (solid, ν/cm^−1^): 1708 s (C=O), 1636 w, 1602 w (C=C_pyridine_), 1147 s, 1036 s (SO_3_). ^1^H-NMR (300.1 MHz, D_2_O, δ, ppm, J/Hz): 1.15 (s, 9H, 3CH_3_), 1.87 (m, 2H, -H_2_C**CH_2_**CH_2_-), 2.91 (t, 2H, ^3^J = 7.1, -CH_2_SO_3_), 3.49 (t, 2H, ^3^J = 7.1, OCH_2_-), 8.42 (d, 1H, ^3^J = 8.1, H3), 8.15 (t, 1H, ^3^J = 8.1, H4), 8.64 (t, 1H, ^3^J = 8.1, H5), 8.85 (d, 1H, ^3^J = 8.1, H6). ^13^C-NMR (75.5 MHz, D_2_O, δ, ppm): 24.51, 26.61, 48.12, 60.24, 74.79, 125.18, 129.69, 142.77, 143.48, 146.96, 162.68. Anal. calcd. for C_18_H_33_N_3_O_9_S_2_: C, 43.27; H, 6.66; N, 8.41; S, 12.83. Found: C, 43.54; H, 6.41; N, 8.70; S, 12.35.

Synthesis of 2-carbamoylpyridine-1-ium-3-cyclohexane hydroxypropane-1-sulfonate (**2f**): Similar to **2b**, using cyclohexanol instead of ethanol, 1.78 g (89%) of compound **2f** · CH_3_CN · H_2_O was obtained, m.p. 173–175 °C (CH_3_CN). IR spectrum (solid, ν/cm^−1^): 1707 s (C=O), 1636 w, 1602 w (C=C_pyridine_), 1178 s, 1147 s, 1037 s (SO_3_). ^1^H-NMR (300.1 MHz, D_2_O, δ, ppm, J/Hz): 1.91 (m, 2H, -H_2_C**CH_2_**CH_2_-), 2.93 (t, 2H, ^3^J = 7.0, -CH_2_SO_3_), 1.17–1.68 (m, 10H, C_6_H_10_), 3.39 (m, 1H, OCH-), 3.63 (t, 2H, ^3^J = 7.0, OCH_2_-), 8.43 (d, 1H, ^3^J = 8.1, H3), 8.16 (t, 1H, ^3^J = 8.1, H4), 8.65 (t, 1H, ^3^J = 8.1, H5), 8.85 (d, 1H, ^3^J = 8.1, H6). ^13^C-NMR (75.5 MHz, D_2_O, δ, ppm): 23.84, 24.78, 25.24, 31.73, 48.02, 65.92, 78.28, 125.13, 129.66, 142.86, 143.58, 146.82, 162.78. Anal. calcd. for C_17_H_29_N_3_O_6_S: C, 50.60; H, 7.24; N, 10.41; S, 7.94. Found: C, 51.01; H, 7.00; N, 10.12; S, 7.41.

Synthesis of 2-(methylcarbamoyl)pyridin-1-ium 3-methoxypropane-1-sulfonate (**2g**): The reaction was performed in 5 mL of methanol. The mixture was heated under reflux for 4 h, and 2.00 g (81% yield) of an oily complex was obtained at m.p. 60–64 °C. IR (solid, ν/cm^−1^): 1679 s (C=O), 1604 (C=C_pyridine_), 1212, 1108, 1032, (SO_3_). ^1^H-NMR (300.1 MHz, D_2_O, ppm, J/Hz): δ 1.92 (m, 2H, C-CH_2_-C), 2.85 (t, 2H, _3_J = 7.1 CH_2_SO_3_), 3.50 (t, 2H, _3_J = 7.3, CH_3_O-CH;_2_), 3.26 (s, 3H, **CH_3_O**-CH_2_), 2.95 (s, 3H, HNCH_3_), 8.12 (t, 1H, _3_J = 7.0, H_5_), 8.81 (d, 1H, _3_J = 8.1, H_6_), 8.34 (d, 1H, _3_J = 6.1, H_4_), 8.63 (t, 1H, _3_J = 7.0, H_5_); ^13^C-NMR (75.5 MHz, D_2_O, ppm): δ 24.03, 26.90, 47.78, 57.61, 70.47, 125.06, 129.57, 143.21, 146.96, 162.39 [10].

Synthesis of 3,3-dimethyl-1-oxo-2,3-dihydro-1H-imidazo [1,5-a]pyridin-4-ium 3-methoxypropane-1-sulfonate (**3a**):(1)A mixture of 0.61 g (0.005 mol) of **1a** and 0.73 g (0.006 mol) of 1,3-propanesultone in 5 mL of methanol was refluxed for 3 h, then 4 mL of acetone was added to the hot solution, and the mixture was refluxed for further 1.5 h. On the next day, the volatiles were removed in a vacuum, the residue was stirred with diethyl ether for 2 h, and the crystals formed were filtered and dried to afford 1.27 g (80%) of compound **3a**, m.p. 174–177 °C (methanol–acetone, 1:20). IR spectrum (solid, ν/cm^−1^): 1729 s (C=O), 1637 w (C=C_pyridine_), 1208 s, 1160 s, 1033 s (SO_3_). ^1^H-NMR (300.1 MHz, D_2_O, δ, ppm, J/Hz): 1.84 (m, 2H, -H_2_C**CH_2_**CH_2_-), 1.89 (s, 6H, 2CH_3_), 2.84 (t, 2H, ^3^J 7.2, -CH_2_SO_3_), 3.25 (s, 3H, ^3^J = 7.2, CH_3_O), 3.47 (t, 2H, ^3^J = 7.2, OCH_2_-), 8.42 (d, 1H, ^3^J = 8.1, H3), 8.31 (t, 1H, ^3^J = 8.1, H4), 8.76 (t, 1H, ^3^J = 8.1, H5), 9.31 (d, 1H, ^3^J = 8.1, H6). ^13^C-NMR (75.5 MHz, D_2_O, δ, ppm): 24.10, 26.48, 47.65, 57.69, 70.57, 84.15, 123.80, 130.93, 138.48, 141.63, 148.03, 158.99. Anal. calcd. for C_13_H_23.5_N_2_O_6.75_S (**3a** ·1.75H_2_O): C, 44.87; H, 6.80; N, 8.05; S, 9.21. Found: C, 44.89; H, 6.18; N, 8.28; S, 10.53.(2)A mixture of 0.37 g (0.003 mol) of **1a** and 0.48 g (0.004 mol) of 1,3-propanesultone in 5 mL of methanol was refluxed for 3 h, then the volatiles were removed in a vacuum, and the remaining crude salt **2a** was dissolved in 10 mL of acetone. On the next day, the volatiles were removed in a vacuum, the residue was stirred with diethyl ether for 2 h, and the crystals formed were filtered and dried to ensure 0.51 g (62%) of unreacted compound **2a** was obtained. By evaporation, 0.11 g (11.7%) of compound **3a** was isolated, m.p. 170–174 °C (methanol–acetone, 1:20). IR spectrum (solid, ν, cm^−1^): 1738 s (C=O), 1633 w (C=C_pyridine_), 1194 s, 1148 s, 1032 s (SO_3_).

Synthesis of 3-ethyl-3-methyl-1-oxo-2,3-dihydro-1H-imidazo [1,5-a]pyridin-4-ium 3-methoxypropane-1-sulfonate (**3b**): Similar to **3a**, using butanone instead of acetone, 1.58 g (91%) of **3b** · 0.5 H_2_O was obtained, m.p. 162–165 °C (acetonitrile–acetone, 1:30). IR spectrum (solid, ν/cm^−1^): 1738 s (C=O), 1633 w (C=C_pyridine_), 1194 s, 1148 s, 1032 s (SO_3_). ^1^H NMR (300.1 MHz, D_2_O, δ, ppm, J/Hz): 0.67 (t, 3H, ^3^J = 7.2, **CH_3_**CH_2_-), 1.95 (m, 2H, -H_2_C**CH_2_**CH_2_-), 2.41 (q, 2H, ^3^J = 7.2, -**CH_2_**CH_3_), 2.89 (t, 2H, ^3^J = 7.2, -CH_2_SO_3_), 3.31 (s, 3H, CH_3_O), 3.53 (t, 2H, ^3^J = 7.2, -OCH_2_), 8.53 (d, 1H, ^3^J = 8.1, H3), 8.42 (t, 1H, ^3^J = 8.1, H4), 8.85 (t, 1H, ^3^J = 8.1, H5), 9.31 (d, 1H, ^3^J = 8.1, H6). ^13^C-NMR (75.5 MHz, D_2_O, δ, ppm): 5.99, 22.97, 26.43, 36.69, 48.03, 57.75, 70.64, 86.95, 123.94, 131.01, 138.54, 141.79, 148.27, 159.44. Anal. calcd. for C_14_H_23_N_2_O_5.5_S: C, 49.54; H, 6.82; N, 8.25; S, 9.44. Found: C, 49.69; H, 6.37; N, 8.27; S, 9.98.

Synthesis of 1-oxo-3,3-dipropyl-2,3-dihydro-1H-imidazo [1,5-a]pyridin-4-ium 3-methoxypropane-1-sulfonate (**3c**): A total of 1.23 g (0.0044 mol) of **2a** was refluxed in a mixture of 4 mL of methanol and 4 mL of dipropyl ketone for 15 h. The volatiles were removed in a vacuum, the residue was stirred with diethyl ether for 2 h, and the crystals formed were filtered and dried to afford 2.28 (75%) of **3c** · 2 HO(CH_2_)_3_SO_3_H · CH_3_CN, m.p. 88–91 °C (CH_3_CN). IR spectrum (solid, ν/cm^−1^): 1733 s (C=O), 1630 w (C=C_pyridine_), 1147 s, 1115 s, 1031 s (SO_3_). ^1^H-NMR (300.1 MHz, D_2_O, δ, ppm, J/Hz): 0.79 (t, 6H, ^3^J = 7.2, 2 **CH_3_**CH_2_CH_2_-), 1.27 (m, 4H, 2 CH_3_CH_2_**CH_2_**-), 1.94 (m, 2H, -CH_2_**CH_2_**CH_2_-), 2.48 (m, 4H, 2 CH_3_**CH_2_**CH_2_-), 2.91 (t, 2H, ^3^J 7.2, CH_2_SO_3_), 3.33 (s, 3H, CH_3_O), 3.55 (t, 2H, ^3^J 7.2, OCH_2_), 8.50 (d, 1H, ^3^J 8.1, H3), 8.40 (t, 1H, ^3^J 8.1, H4), 8.80 (t, 1H, ^3^J 8.1, H5), 9.24 (d, 1H, ^3^J 8.1, H6). ^13^C-NMR (75.5 MHz, D_2_O, δ, ppm): 12.59, 15.10, 24.17, 40.23, 47.94, 57.75, 70.65, 89.11, 124.10, 131.38, 138.58, 148.43, 159.10. Anal. calcd. for C_25_H_47_N_3_O_13_S_3_: C, 43.27; H, 6.82; N, 6.05; S, 13.86. Found: C, 42.54; H, 6.64; N, 6.46; S, 13.20.

Synthesis of 1′-oxo-1′,2′-dihydrospiro[cyclopentane-1,3′-imidazo [1,5-a]pyridin]-4′-ium 3-methoxypropane-1-sulfonate (**3d**): Similar to **3a**, using cyclopentanone instead of acetone, 1.60 g (85%) of **3d** · 2 H_2_O was obtained, m.p. 177–179 °C (acetonitrile–acetone, 1:3). IR spectrum (solid, ν/cm^−1^): 1726 s (C=O), 1637 w (C=C_pyridine_), 1169 s, 1112 s, 1039 s (SO_3_). ^1^H-NMR (300.1 MHz, D_2_O, δ, ppm, J/Hz): 1.88–2.55 (m, 8H cycle; 2H, -H_2_C**CH_2_**CH_2_-), 2.89 (t, 2H, ^3^J = 7.2, CH_2_SO_3_), 3.29 (s, 3H, CH_3_O), 3.49 (t, 2H, ^3^J = 7.2, -OCH_2_), 8.45 (d, 1H, ^3^J = 8.1, H3), 8.35 (t, 1H, ^3^J = 8.1, H4), 8.78 (t, 1H, ^3^J = 8.1, H5), 9.30 (d, 1H, ^3^J = 8.1, H6). ^13^C-NMR (75.5 MHz, D_2_O, δ, ppm): 22.80, 24.16, 40.33, 47.93, 57.75, 70.64, 92.40, 123.29, 131.00, 138.51, 142.18, 148.11, 159.20. Anal. calcd. for C_15_H_26_N_2_O_7_S. Calculated: C, 47.61; H, 6.92; N, 7.40; S, 8.47. Found: C, 47.12; H, 6.65; N, 7.87; S, 8.79.

Synthesis of 1′-oxo-1′,2′-dihydrospiro[cyclohexane-1,3′-imidazo [1,5-a]pyridin]-4′-ium 3-methoxypropane-1-sulfonate (**3e**): Similar to **3a**, using cyclohexanone instead of acetone, 1.71 g (94%) of compound **3e** · 0.5 H_2_O was obtained, m.p. 213–216 °C (CH_3_CN). IR spectrum (solid, ν/cm^−1^): 1735 s (C=O), 1638 w (C=C_pyridine_), 1169 s, 1114 s, 1036 s (SO_3_). ^1^H-NMR (300.1 MHz, D_2_O, δ, ppm, J/Hz): 1.4–2.3 (m, 10H cycle; 2H -H_2_C**CH_2_**CH_2_-), 2.90 (t, 2H, ^3^J = 7.2, -CH_2_SO_3_), 3.29 (s, 3H, CH_3_O), 3.53 (t, 2H, ^3^J = 7.2, OCH_2_-), 8.48 (d, 1H, ^3^J 8.1, H3), 8.34 (t, 1H, ^3^J 8.1, H4), 8.81 (t, 1H, ^3^J 8.1, H5), 9.33 (d, 1H, ^3^J 8.1, H6). ^13^C-NMR (75.5 MHz, D_2_O, δ, ppm): 22.57, 22.98, 23.17, 24.17, 47.94, 57.75, 70.65, 86.88, 123.87, 130.84, 138.67, 141.48, 148.11, 159.64. Anal. calcd. for C_16_H_25_N_2_O_5.5_S: C, 52.58; H, 6.89; N, 7.66; S, 8.72. Found: C, 52.49; H, 6.65; N, 8.02; S, 8.96.

Synthesis of 3,3-dimethyl-1-oxo-2,3-dihydro-1H-imidazo [1,5-a]pyridin-4-ium 3-ethoxypropane-1-sulfonate (**3f**): Similar to **3a**, using ethanol instead of methanol, 2.05 g (82%) of **3f** · HO(CH_2_)_3_SO_3_H · CH_3_CN was obtained, m.p. 118–121 °C (CH_3_CN). IR spectrum (solid, ν/cm^−1^): 1732 s (C=O), 1635 w (C=C_pyridine_), 1156 s, 1123 s, 1034 s (SO_3_). ^1^H-NMR (300.1 MHz, D_2_O, δ, ppm, J/Hz): 1.10 (t, 3H, ^3^J = 6.9, **CH_3_**CH_2_), 1.89 (s, 6H, 2CH_3_), 1.94 (m, 2H, -H_2_C**CH_2_**CH_2_-), 2.91 (t, 2H, ^3^J = 7.0, -CH_2_SO_3_), 3.49 (t, 2H, ^3^J = 7.0, OCH_2_-), 3.63 (q, 2H, ^3^J = 6.9, CH_3_**CH_2_**-), 8.45 (d, 1H, ^3^J = 8.1, H3), 8.39 (t, 1H, ^3^J = 8.1, H4), 8.79 (t, 1H, ^3^J = 8.1, H5), 9.32 (d, 1H, ^3^J = 8.1, H6). ^13^C-NMR (75.5 MHz, D_2_O, δ, ppm): 14.18, 21.17, 24.35, 26.61, 48.02, 68.52, 84.21, 122.81, 131.02, 138.54, 141.64, 148.12, 159.01. Anal. calcd. for C_19_H_33_N_3_O_9_S_2_. Calculated: C, 44.60; H, 6.50; N, 8.21; S, 12.53. Found: C, 44.09; H, 6.46; N, 8.38; S, 11.79.

Synthesis of 3,3-dimethyl-1-oxo-2,3-dihydro-1H-imidazo [1,5-a]pyridin-4-ium 3-isopropoxypropane-1-sulfonate (**3g**): Similar to **3a**, using isopropanol instead of methanol, 1.12 g (43%) of **3g** · HO(CH_2_)_3_SO_3_H · CH_3_CN was obtained, m.p. 139–140 °C (CH_3_CN). IR spectrum (solid, ν/cm^−1^): 1734 s (C=O), 1638 w (C=C_pyridine_), 1156 s, 1123 s, 1033 s (SO_3_). ^1^H-NMR (300.1 MHz, D_2_O, δ, ppm, J/Hz): 0.86 (d, 6H, ^3^J = 7.0, 2CH_3_), 1.93 (m, 1H, CHO), 1.90 (m, 2H, -H_2_C**CH_2_**CH_2_-), 1.94 (s, 6H, 2CH_3_), 2.89 (t, 2H, ^3^J = 7.0, -CH_2_SO_3_), 3.54 (t, 2H, ^3^J = 7.0, OCH_2_-), 8.48 (d, 1H, ^3^J = 8.1, H3), 8.37 (t, 1H, ^3^J = 8.1, H4), 8.79 (t, 1H, ^3^J = 8.1, H5), 9.34 (d, 1H, ^3^J = 8.1, H6). ^13^C-NMR (75.5 MHz, D_2_O, δ, ppm): 18.63, 24.34, 26.66, 27.59, 48.06, 68.85, 77.51, 84.24, 122.83, 131.06, 138.57, 141.63, 148.16, 158.99. Anal. calcd. for C_20_H_35_N_3_O_9_S_2_. Calculated: C, 45.69; H, 6.71; N, 7.99; S, 12.20. Found: C, 46.35; H, 6.35; N, 7.75; S, 12.75.

Synthesis of 3,3-dimethyl-1-oxo-2,3-dihydro-1H-imidazo [1,5-a]pyridin-4-ium 3-isobutoxixipropane-1-sulfonate (**3h**): Similar to **3a**, using isobutanol instead of methanol, 1.96 g (73%) of **3h** · HO(CH_2_)_3_SO_3_H · CH_3_CN was obtained, m.p. 147–150 °C (CH_3_CN). IR spectrum (solid, ν/cm^−1^): 1740 s (C=O), 1632 w (C=C_pyridine_), 1149 s, 1031 s (SO_3_). ^1^H-NMR (300.1 MHz, D_2_O, δ, ppm, J/Hz): 0.85 (d, 6H, ^3^J 7.0, (**CH_3_**)_2_CH-), 2.02 (m, 6H, 2CH_3_), 1.87 (m, 1H, (CH_3_)_2_**CH**-)), 1.98 (s, 6H, 2CH_3_), 2.94 (t, 2H, ^3^J = 7.0, **CH_2_**SO_3_), 3.15 (d, 2H, ^3^J = 7.0, CH**CH_2_**O), 3.64 (t, 2H, ^3^J = 7.0, OCH_2_-), 8.52 (d, 1H, ^3^J = 8.1, H3), 8.43 (t, 1H, ^3^J = 8.1, H4), 8.85 (t, 1H, ^3^J = 8.1, H5), 9.38 (d, 1H, ^3^J = 8.1, H6). ^13^C-NMR (75.5 MHz, D_2_O, δ, ppm): 18.63, 24.34, 26.66, 27.59, 48.06, 60.27, 68.85, 77.51, 84.24, 122.83, 131.06, 138.57, 141.63, 148.16, 158.99. Anal. calcd. for C_21_H_37_N_3_O_9_S_2_. Calculated (%): C, 46.74; H, 6.91; N, 7.79; S, 11.88. Found: C, 46.25; H, 6.81; N, 8.01; S, 11.29.

Synthesis of 3,3-dimethyl-1-oxo-2,3-dihydro-1H-imidazo [1,5-a]pyridin-4-ium 3-tretbutoxipropane-1-sulfonate (**3i**): Similar to **3a**, using tert-butanol instead of methanol, 0.45 g (23%) of **3i** · 2 H_2_O was obtained, m.p. 118–121 °C. IR spectrum (solid, ν/cm^−1^): 1735 s (C=O), 1638 w (C=C_pyridine_), 1152 s, 1033 s (SO_3_). ^1^H-NMR (300.1 MHz, D_2_O, δ, ppm, J/Hz): 1.14 (br. s, 9H, 3CH_3_), 1.86–2.13 (m, 2H, -H_2_C**CH_2_**CH_2_-; s, 6H, 2CH_3_), 2.93 (br. m, 2H, -CH_2_SO_3_), 3.52 (br. m, 2H, OCH_2_-), 8.53 (br. m, 1H, H3), 8.44 (br. m, H4), 8.89 (br. m, 1H, H5), 9.42 (br. m, 1H, H6). ^13^C-NMR (75.5 MHz, D_2_O, δ, ppm): 15.63, 23.19, 24.54, 26.64, 31.75, 48.14, 69.83, 74.80, 85.00, 123.91, 131.04, 138.55, 142.77, 148.14, 162.28. Anal. calcd. for C_16_H_30_N_2_O_7_S. Calculated: C, 48.71; H, 7.66; N, 7.10; S, 8.12. Found: C, 48.53; H, 7.30; N, 7.50; S, 8.81.

Synthesis of 3,3-dimethyl-1-oxo-2,3-dihydro-1H-imidazo [1,5-a]pyridin-4-ium 3-cyclohexaneoxipropane-1-sulfonate (**3j**): Similar to **3a**, using cyclohexanol at 78–80 °C instead of methanol at reflux, 1.68 g (80%) of **3j** · 2 H_2_O was obtained, m.p. 167–170 °C (CH_3_CN). IR spectrum (solid, ν/cm^−1^): 1735 s (C=O), 1632 w (C=C_pyridine_), 1158 s, 1037 s (SO_3_). ^1^H-NMR (300.1 MHz, D_2_O, δ, ppm, J/Hz): 1.91 (m, 2H, -CH_2_**CH_2_**CH_2_-), 1.95 (s, 6H, 2CH_3_), 2.93 (t, 2H, ^3^J = 7.0, -CH_2_SO_3_), 1.17–1.68 (m, 10H, C_6_H_10_), 3.39 (m, 1H, OCH), 3.63 (t, 2H, ^3^J = 7.0, OCH_2_-), 8.51 (d, 1H, ^3^J = 7.2, H3), 8.43 (t, 1H, ^3^J = 7.2, H4), 8.85 (t, 1H, ^3^J = 7.2, H5), 9.38 (d, 1H, ^3^J = 7.2, H6). ^13^C-NMR (75.5 MHz, D_2_O, δ, ppm): 23.87, 24.79, 25.26, 26.62, 31.75, 48.14, 65.94, 78.32, 84.23, 122.54, 123.90, 131.03, 138.54, 148.14, 158.50. Anal. calcd. for C_18_H_32_N_2_O_7_S. Calculated: C, 51.41; H, 7.67; N, 6.66; S, 7.62. Found: C, 51.02; H, 7.42; N, 7.03; S, 8.10.

Reactions of hydrolysis of compounds **3a,d,e**

Synthesis of 2-carbamoylpyridine-1-ium-3-methoxypropane-1-sulfonate (**3a**). 

(a)0.11 g (0.003 mol) of **3a** in 3 mL of water was stirred for 7 days at room temperature. After evaporation, 0.10 g (91%) of **3a** was obtained.(b)0.36 g (0.011 mmol) of **3a** in 4 mL of water was refluxed for 4 h. After evaporation and recrystallisation of the residue from CH_3_CN, 0.10 g (32%) of **2a** was obtained.(c)0.31 g (0.0097 mol) of **3a** in 4 mL of water was stirred at 70–80 °C for 5 h. After evaporation, 0.17 g (63%) of **2a** was obtained.(d)0.11 g (0.003 mol) of **3d** in 4 mL of water was stirred at 70–80 °C for 5 h. After evaporation, 0.08 g of a mixture of **3a** and **3d** was obtained.(e)0.11 g (0.003 mol) of **3e** in 4 mL of water was stirred at 70–80 °C for 5 h. After evaporation, 0.08 g (97%) of **3a** was obtained.

### 3.2. Calculation Details

Quantum-chemical calculations were carried out using Gaussian software, ver. 09 rev. C01 [36] and visualised using ChemCraft software, ver. 1.8 [37]. All geometric and energy-related values were obtained using the M052X hybrid functional [38] with empirical dispersion [39] and the TZVP basis set [40]. Chemical shifts were calculated using the continuous set of gauge transformations (CSGT) method [40,41,42]. The correlation coefficients for experimental and calculated chemical shifts in NMR spectra were around 99% (Appendix A).

The above method was used for the full optimisation of the structures of reactants and products (Appendix A). The calculations were carried out in the approximation of isolated molecules. The solvent effects were taken into account using the integral equation formalism variant of the polarisable continuum model (IEFPCM).

The correspondence of the calculated structures to minima on the potential energy surface was assessed by the absence of negative elements in the diagonalised Hessian matrix. The transition states were identified by the presence of a single negative element in the matrix.

Thermal effects of reactions and activation enthalpies were calculated as the difference between the absolute enthalpies of the final (or transition) and initial states of the process. Absolute enthalpies were calculated as the sum of total energy, zero-point energy and thermal correction for the enthalpy change from zero to 298 K. The latter values were obtained by frequency calculations using common equations of statistical thermodynamics.

The mutual arrangement of the protonated picolinamide cation and the sulfonate anion was determined using electrostatic potential (ESP) maps, which were calculated using MultiWFN software, ver. 3.8 [43] and visualised using VMD software (version 1.1) [44].

The structure of the pre-reaction complex was determined using molecular dynamics (MD) modelling of a system containing a protonated picolinamide cation, a sulfonate anion, a molecule of acetone and 2000 solvent (methanol) molecules (Appendix A). MD modelling was performed for a cube-shaped system with periodic boundary conditions (PBC) using the OPLS4 force field [45].

The simulation of the isobaric-isothermal process was carried out using the NPT molecular ensemble. The dynamics simulation was recorded for 10 ns at 337 K (the boiling point of methanol). A total of 5000 frames were used for the statistical analysis. The analysis of the MD trajectory and the construction of volumetric maps for the PBC space were carried out using VMD software [44].

### 3.3. X-ray Crystallographic Studies

Single-crystal X-ray studies of compounds **2a** and **3a** were carried out in the Center for Molecule Composition Studies of INEOS RAS using APEX3 software [46]. The data obtained were then integrated with SAINT. SADABS was used for scaling, empirical absorption corrections and generation of data files for structure solution and refinement.

The structures were solved by a dual-space algorithm and refined in anisotropic approximation for non-hydrogen atoms against F^2^(hkl). The positions of hydrogen atoms in methyl, methylene and aromatic fragments were calculated for idealised geometry and refined with constraints applied to C–H and N–H bond lengths and equivalent displacement parameters (U_eq_(H) = 1.2U_eq_(X) for XH_2_ groups and U_eq_(H) = 1.5U_eq_(Y) for YH_3_ groups). All structures were solved using ShelXT [47] and refined using ShelXL software [48]. Molecular graphics were drawn using OLEX2 software [49]. Structure **2a** was refined as a two-component non-merohedral twin using PLATON software [50]. The scale factors for the twin components were 0.880(4) and 0.120(4). Structure **3a** was refined as a two-component non-merohedral twin using the TWINABS program implemented in APEX3 software [46].

The supplementary crystallographic data for **2a** and **3a** (2287812 and 2287813) are available free of charge from the Cambridge Crystallographic Data Centre at https://www.ccdc.cam.ac.uk/structures (accessed on 9 August 2023).

## 4. Conclusions

In contrast to meta- and para-pyridinecarboxamides, their ortho-analogues (picolinamides) react with 1,3-propanesultone in hot methanol to give pyridinium salts with a protonated endocyclic nitrogen atom [10]. In this work, the scope of this reaction has been extended to other alcohols. In the presence of ketones, the reaction products **2a**–**f** form a new type of heterocyclic salts **3a**–**j** containing an imidazolidin-4-one ring. According to the X-ray study, the main structural parameters of compounds **2a** and **3a** are typical for pyridinium salts of alkylsulfonic acids. The calculated thermodynamic parameters of reactions leading to the formation of **2a** and **3a** correlate with the size of alkyl substituents in substrates and reaction yields. Using **3a** as an example, the potential biological activity of imidazolidin-4-one derivatives has been evaluated in our recent work [11]. The series of synthesised salts **3** can be used to study the effects of various structural features on their drug-like activity and identify potential lead compounds. In addition, further studies of the reaction mechanism will allow us to apply the proposed synthetic route to a broader range of carbonyl compounds.

## Data Availability

All spectra and XRD data are available from the authors.

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
