# Peer review of "Reaction of Picolinamides with Ketones Producing a New Type of Heterocyclic Salts with an Imidazolidin-4-One Ring"

_molecules, 2023, doi:10.3390/molecules29010206_

Round 1

Reviewer 1 Report

Comments and Suggestions for Authors

the manuscript entitled "Reaction of Picolinamides with Ketones Producing a New Type of Heterocyclic Salts With a Imidazoline Ring" deals with the formation of new heterocyclic salts based on a pyridine scaffold. It's a nice paper, discussion and conclusions are supported by consistent data, anyway some corrections have to be performed and a bigger attention to the introduction and the conclusions has to be paid. Then, major revisions are required. My comments:

Title: “With”. Lowercase instead of uppercase.

Introduction and Abstract: Emphasis should be added to the reasons behind the choice to investigate the topic.

Scheme 1: Define R1 and R2

Line 50: What happens when DMSO is used as medium reaction? DMSO is a strong hydrogen bonding competitor and no intramolecular bonding between amido group and the endocyclic nitrogen atom should occur.

Lines 56-57:” 3a could be isolated as individual compound and characterised by multinuclear (1H, 13C and 15N) and 2D NMR spectroscopy”. The 15N NMR spectrum of this compound is not present in the Supplementary part. Mention to 15N NMR spectrum could also be avoided, or the 15N NMR spectrum (and characterization data) should be added.

Line 79: Which is the main product in this case?

Line 82: Report the structure of the emiketal intermediate in a new Scheme. Have been performed further investigations about the mechanism?

Lines 93-97: In which sense the separation is problematic? How has 3i been separated from 2e? There isn’t a specific protocol for this compound in the 3.1.1 paragraph in the main text.

Lines 98-99: It appears a little strange that the salt doesn’t react with benzaldehyde. For tert-butyl-methyl-ketone sterical hindrance could be a reason. Which hypothesis has been formulated for its different reactivity? Have other aldehydes been tested?

Conclusions are poorly described. For example, which are the future perspectives of this work?

Author Response

Dear Editor,

The authors are grateful to the reviewer for their interest to our work. Their comments have helped us not only to improve the paper but also develop a plan for further studies. Our replies to reviewers' comments are given below.

Answers to reviewer #1

  1. Title: “With”. Lowercase instead of uppercase.

Answer: done.

  1. Introduction and Abstract: Emphasis should be added to the reasons behind the choice to investigate the topic.

Answer: thank you for the important point, done.

  1. Scheme 1: Define R1 and R2

Answer: done.

  1. Line 50: What happens when DMSO is used as medium reaction? DMSO is a strong hydrogen bonding competitor and no intramolecular bonding between amido group and the endocyclic nitrogen atom should occur.

Answer: indeed, the use of DMSO as a solvent could potentially change the selectivity and predominant direction of the substitution. These studies would require some trials for determining the optimum reaction conditions. We are going to investigate this in the future.

  1. Lines 56-57:” 3a could be isolated as individual compound and characterized by multinuclear (1H, 13C and 15N) and 2D NMR spectroscopy”. The 15N NMR spectrum of this compound is not present in the Supplementary part. Mention to 15N NMR spectrum could also be avoided, or the 15N NMR spectrum (and characterization data) should be added.

Answer: we are sorry for this technical error. We have removed the reference to the 15N NMR spectrum of 3a, as 15N NMR studies were performed only for a mixture of compounds 2a and 3a. The spectra will be supplied along with other corrections.

  1. Line 79: Which is the main product in this case?

Answer: the main product was unreacted compound 2a. This is now stated clearly in the experimental section (lines 321–326) as follows:

A mixture of 0.37 g (0.003 mol) of 1a and 0.748 g (0.004 mol) of 1,3-propanesultone in 5 mL of methanol was refluxed for 3 h. The solvent was removed in vacuum, and the remaining crude salt 2a (0.82 g, 0.003 mol) was dissolved in 10 mL of acetone. Next day, the volatiles were removed in vacuum, the residue was stirred with diethyl ether for 2 h, and the crystals formed were filtered and dried to afford 0.51 g (62%) of unreacted compound 2a. The evaporation of the remaining solution afforded 0.11 g (12%) of compound 3a, m.p. 170–174 °C (methanol–acetone, 1:20). IR spectrum (solid, ν, cm–1): 1738 s (C=O), 1633 w (C=Cpyridine), 1194 s, 1148 s, 1032 s (SO3).

  1. Line 82: Report the structure of the semiketal intermediate in a new Scheme. Have been performed further investigations about the mechanism?

Answer: Scheme 4 is now added. No further studies of the mechanism have been performed.

  1. Lines 93–97: In which sense the separation is problematic? How has 3i been separated from 2e? There isn’t a specific protocol for this compound in the 3.1.1 paragraph in the main text.

Answer: after the evaporation and filtration of the reaction mixture, product 3i was washed repeatedly with ether until the IR absorption of ν(C=O) in 2e (strong band at 1708 cm–1) disappeared. The purity of 3i with the IR absorption of ν(C=O) at 1708 cm–1 was confirmed NMR (1H and 13C) and elemental analysis.

  1. Lines 98–99: It appears a little strange that the salt doesn’t react with benzaldehyde. For tert-butyl-methyl-ketone sterical hindrance could be a reason. Which hypothesis has been formulated for its different reactivity? Have other aldehydes been tested?

Answer: the use of other aldehydes is a part of our plans for future studies. At the moment, sterical hindrance and the resonance effect of the benzene ring seem to be the most likely reasons for the low reactivity of benzaldehyde.

  1. Conclusions are poorly described. For example, which are the future perspectives of this work?

Answer: the Conclusions section has been revised.

Reviewer 2 Report

Comments and Suggestions for Authors

Authors present a reaction leading to a set of heterocyclic organic salts formed from picolinamides. This study is the continuation of their preious work. There is synthesis, computation, X-ray, so the manuscript is quite comprehensive but contains a lot of questionable part. 

1. In the SuppInfo the NMR spectra should present the range of 0-12 and 0-220 in case of 1H NMR and 13C NMR, respectively.

2. In the MD simulation the picolinamide should be in the center of the grid.

3. All Figures and schemes in the manuscript should have a caption.

4. Scheme 1 present only a single route, not "routes" as in the caption. 

5. Introduction is very short. Some paragraphs from later could be transferred, and the significance should be highlighted. 

6. For compond B NHR' is indicated, but based on later results, it might be better to indicate NH2.

7. AUthors write imidazoline, imidazolinium, but the compounds contain an oxo group. The names of the rings should be corrected.

8. There is a scheme showing the two-step synthesis of so-called imidazolines (to be corrected). Next, in the paragaph there are more synthetic options that might be shown as well on a scheme. 

9. Tables should have a reaction scheme over them containing also the reaction conditions. 

10. The crystal structure seems to show the cationic ring system flat (might be only from this point of view).Authors are asked to analyze the 2D-3D structure of the ring system that also contains an sp3 carbon, so being flat would be surprising. 

11. What is the rationale for testing alcohol stability? Are drug substances tested in boiling methanol? If authors know the limited stability in alcohols, why did they boil with the ketones in that solvent and why did not chose a different one? Were solvents optimized?

12. For trnsition states the # sign should be added. Transition state barrier energy/enthalpy/Gibbs free energy would be more characteristic than the thermodinamic data, please include. 

13. Benzene was used as a crystallization solvent. It is suggested to change to a less carcinogenic solvent. 

14. "mmol" and "mol" should be checked in the experimental.

Comments on the Quality of English Language

-

Author Response

Dear Editor,

The authors are grateful to the reviewer for their interest to our work. Their comments have helped us not only to improve the paper but also develop a plan for further studies. Our replies to reviewers' comments are given below.

Answers to reviewer

1. In the SuppInfo the NMR spectra should present the range of 0-12 and 0-220 in case of 1H NMR and 13C NMR, respectively.

Answer: done.

2. In the MD simulation the picolinamide should be in the center of the grid.

Answer: figure S1 shows the initial state of the MD system. The exact locations of molecules are not important, as main purpose of this figure was to outline the scale and complexity of the model. The software used allows to centre the picolinamide and carry out statistical analysis with respect to that particular molecule, which was indeed done in this study.

3. All Figures and schemes in the manuscript should have a caption.

Answer: done.

4. Scheme 1 present only a single route, not "routes" as in the caption.

Answer: done.

5. Introduction is very short. Some paragraphs from later could be transferred, and the significance should be highlighted.

Answer: done.

6. For compound B NHR' is indicated, but based on later results, it might be better to indicate NH2.

Answer: the structure of type B has also been reported for a NHMe-derivative of picolinamide [Kramarova, E. P., Borisevich, S. S., Khamitov, E. M., Korlyukov, A. A., Dorovatovskii, P. V., Shagina, A. D., Mineev, K. S., Tarasenko, D. V., Novikov, R. A., Lagunin, A. A., Boldyrev, I., Ezdoglian, A. A., Karpechenko, N. Y., Shmigol, T. A., Baukov, Y. I., and Negrebetsky, V. V. Pyridine Carboxamides Based on Sulfobetaines: Design, Reactivity, and Biological Activity. Molecules 2022, 27, 7542.]

7. Authors write imidazoline, imidazolinium, but the compounds contain an oxo group. The names of the rings should be corrected.

Answer: thank you, this technical error has been corrected.

8. There is a scheme showing the two-step synthesis of so-called imidazolines (to be corrected). Next, in the paragaph there are more synthetic options that might be shown as well on a scheme.

Answer: thank you for this suggestion. Unfortunately, we cannot implement it because the formation of salts 3requires the presence of an alcohol. Therefore, the isolation of intermediate salts 2 becomes very problematic, so only the first synthetic route is feasible. However, we have added Scheme 4 that describes the possible mechanism of the reaction.

9. Tables should have a reaction scheme over them containing also the reaction conditions.

Answer: done.

10. The crystal structure seems to show the cationic ring system flat (might be only from this point of view). Authors are asked to analyze the 2D-3D structure of the ring system that also contains an sp3 carbon, so being flat would be surprising.

Answer:

11. What is the rationale for testing alcohol stability? Are drug substances tested in boiling methanol? If authors know the limited stability in alcohols, why did they boil with the ketones in that solvent and why did not chose a different one? Were solvents optimized?

Answer: since the reaction is carried out in an alcohol, trace amounts of the solvent can be present in the compound used in pre-clinical studies. Therefore, it was necessary to study the effect of methanol on the stability of the final product.

12. For transition states the # sign should be added. Transition state barrier energy/enthalpy/Gibbs free energy would be more characteristic than the thermodynamic data, please include.

Answer: the "#" symbol has been added. The modelling of transition states for the whole spectrum of studied compounds is a complicated and time-consuming process, so it was not carried out in this study. However, we will take this into account when preparing further publications.

13. Benzene was used as a crystallization solvent. It is suggested to change to a less carcinogenic solvent.

Answer: the recrystallisation can be carried out using acetonitrile only. This is now reflected in the Materials and Methods section of the updated manuscript.

14. "mmol" and "mol" should be checked in the experimental.

Answer: done.

Round 2

Reviewer 1 Report

Comments and Suggestions for Authors

All the comments to the manuscript have been fully addressed in the revised article. Now, the paper satisfies all the requisites for the publication on Molecules, so I recommend to accept it in the present form.

Author Response

Many thanks for your attention

Reviewer 2 Report

Comments and Suggestions for Authors

Authors answered almost all the questions and comments by the Referees. The manuscript was improved, however, there are still open questions that in my view should be corrected. 

1. Scheme 3 caption still suggests synthetic routes, but there is only a single route presented. In the text harsch conditions are mentioned, but not prensented in details. It is still suggested to show other mentioned synthetic routes as well to highlight the advantages of the methodology in the manuscript. 

2. Question 10 was not answered: 10. The crystal structure seems to show the cationic ring system flat (might be only from this point of view). Authors are asked to analyze the 2D-3D structure of the ring system that also contains an sp3 carbon, so being flat would be surprising.

3. Tables still do not contain corresponding reaction schemes and conditions. 

4. The computation of at least a representative transition state would be necessary for this journal. 

5. I agree that alcohol stability might be important, but it still does not seem rational to perform the test in boiling methanol. 

Author Response

Dear Editor, my co-authors and I once again thank the reviewers for their attentive and interested reading of our article. The comments made are very valuable and serve to improve the quality of the article. Below are our answers to them.

  1. A) Scheme 3 caption still suggests synthetic routes, but there is only a single route presented.

Answer: Fixed on «General synthetic route to compounds 2a–g and 3a–j”. Similar changes have also been made to the description of Scheme 1 «General synthetic route to compound 

  1. B) In the text harsch conditions are mentioned, but not prensented in details. It is still suggested to show other mentioned synthetic routes as well to highlight the advantages of the methodology in the manuscript.

Answer: The discussion of the advantages of our synthetic approaches in comparison to other synthetic routes is not applicable because in our work, we report new types of imidazolidin-4-ones with novel organic anions, not just a new synthetic route to known compounds. Unfortunately, it was not clear from the original version of the paper, so we have now made some changes in lines 60, 61, 70 and 71 of the latest manuscript, as follows.

Lines 60 and 61:

At present, imidazolidin-4-ones are usually prepared by multistage synthetic processes [15-23] (Scheme 3).

Lines 70 and 71:

In this study, the synthesis, structure and properties of new types of imidazolidin-4-ones with novel organic anions are reported.

  1. Question 10 was not answered: 10. The crystal structure seems to show the cationic ring system flat (might be only from this point of view). Authors are asked to analyze the 2D-3D structure of the ring system that also contains an sp3 carbon, so being flat would be surprising.

Answer: We answered this question earlier (see “answers.docx”, No. 20):

Question 20: The crystal structure seems to show the cationic ring system flat (might be only from this point of view). Authors are asked to analyze the 2D-3D structure of the ring system that also contains an sp3 carbon, so being flat would be surprising.

Answer: Indeed, this five-membered ring is planarised, but it is not entirely planar. The deviation of C3 from the plane of the four other atoms is only 0.06 А. The planarisation of the five-membered ring is likely to be caused by the C=O bond. In CSD, there are no exact matches for this cation, although structure YUDLOZ (10.1016/j.inoche.2009.06.024 ) can be considered as the closest analogue.

  1. Tables still do not contain corresponding reaction schemes and conditions. 

Answer:

  1. In accordance with the recommendations of the Reviewer, Scheme 3 was replaced, which did not fully reflect the data presented in Tables 1 and 2:
  2. The conditions of the reactions are described in detail in the Experimental Part.

Under similar conditions, these reactions produced a broad range of pyridinium (2a-g) and imidazolidin-4-ones (3a-j) salts (Scheme 1, Tables 1 and 2, see also Experimental Part).

  1. The computation of at least a representative transition state would be necessary for this journal. 

Answer:  The transition state is shown on page 9, line 190.

  1. I agree that alcohol stability might be important, but it still does not seem rational to perform the test in boiling methanol. 

Answer: In accordance with the recommendations of the reviewer, the results of studies on the stability of synthesized compounds in boiling methanol have been removed from the relevant section.
